# Tailored Melatonin- and Donepezil-Based Hybrids Targeting Pathognomonic Changes in Alzheimer’s Disease: An In Vitro and In Vivo Investigation

**DOI:** 10.3390/ijms25115969

**Published:** 2024-05-29

**Authors:** Rositsa Mihaylova, Violina T. Angelova, Jana Tchekalarova, Dimitrinka Atanasova, Petja Ivanova, Rumyana Simeonova

**Affiliations:** 1Department “Pharmacology, Pharmacotherapy and Toxicology”, Faculty of Pharmacy, Medical University of Sofia, 1431 Sofia, Bulgaria; rmihaylova@pharmfac.mu-sofia.bg (R.M.); V.stoyanova@pharmfac.mu-sofia.bg (V.T.A.); rsimeonova@pharmfac.mu-sofia.bg (R.S.); 2Institute of Neurobiology, Bulgarian Academy of Sciences, 1113 Sofia, Bulgaria; didiatanasova7@gmail.com (D.A.); Ivanova.petya91@gmail.com (P.I.); 3Department of Anatomy, Faculty of Medicine, Trakia University, 6003 Stara Zagora, Bulgaria

**Keywords:** Alzheimer’s disease, neurodegenerative disorders, β-amyloid, antioxidant, anticholinesterase, neuroprotective activity, melatonin

## Abstract

A plethora of pathophysiological events have been shown to play a synergistic role in neurodegeneration, revealing multiple potential targets for the pharmacological modulation of Alzheimer’s disease (AD). In continuation to our previous work on new indole- and/or donepezil-based hybrids as neuroprotective agents, the present study reports on the beneficial effects of lead compounds of the series on key pathognomonic features of AD in both cellular and in vivo models. An enzyme-linked immunosorbent assay (ELISA) was used to evaluate the anti-fibrillogenic properties of 15 selected derivatives and identify quantitative changes in the formation of neurotoxic β-amyloid (Aβ42) species in human neuronal cells in response to treatment. Among the most promising compounds were **3a** and **3c**, which have recently shown excellent antioxidant and anticholinesterase activities, and, therefore, have been subjected to further in vivo investigation in mice. An acute toxicity study was performed after intraperitoneal (i.p.) administration of both compounds, and 1/10 of the LD_50_ (35 mg/kg) was selected for subacute treatment (14 days) with scopolamine in mice. Donepezil (DNPZ) and/or galantamine (GAL) were used as reference drugs, aiming to establish any pharmacological superiority of the multifaceted approach in battling hallmark features of neurodegeneration. Our promising results give first insights into emerging disease-modifying strategies to combine multiple synergistic activities in a single molecule.

## 1. Introduction

Alzheimer’s disease (AD) is the most common form of dementia affecting more than 10% of the adult population worldwide. As a neurodegenerative disease, it is characterized by progressive brain atrophy and neuronal loss, resulting in an evolving neurological, cognitive, behavioral, and functional symptomatology that progressively worsens over time and is often first diagnosed at a very advanced stage [1,2]. With no curative treatment available to date, it has been an ongoing goal of leading research and clinical experts to delay the progression of the disease by targeting some of the well-established neuromolecular mechanisms and predisposing factors for its development [1,3]. A plethora of pathophysiological events have been shown to play a synergistic role in the etiopathogenesis of AD, including premature synaptotoxicity, changes in neurotransmitter expression, aberrant cholinergic transmission, oxidative stress-induced glial cell activation, mitochondrial dysfunction, neuroinflammation, and neurophil loss [4,5,6,7]. In addition, key pathognomonic changes in the histopathology of the AD brain have been identified and have served as cornerstones in building the main hypotheses for the disease’s origin. It has been shown that intraneuronal aggregation and deposition of neurofibrillary tangles (NFTs) composed of hyperphosphorylated tau protein, coupled with excessive extracellular β-amyloid (Aβ) plaque formation induce neurotoxicity, oxidative stress, inflammation, and, ultimately, lead to neuronal apoptosis and synapse loss [1,7,8].

For almost 30 years, the primary hypothesis for AD has been based on the premise of a disrupted Aβ homeostasis as a result of an imbalance between Aβ peptide production and their clearance in the brains of AD patients [9,10,11]. The Aβ is derived from a precursor amyloid protein (APP), which undergoes a cascade of degradation via either the β- or α-metabolic pathway. Of particular relevance to AD is the amyloidogenic β-secretase pathway, in which APP is cleaved to produce a soluble beta-sAPP and a C-terminal fragment (CTF) of 99 amino acids (C99). C99 is then further cleaved by a γ-secretase isoenzyme to yield the neurotoxic Aβ40/42 peptide fragments, found to be the main components of the senile plaques and a histological hallmark of AD [5,9,11,12,13].

Dysfunction of the cholinergic system has also been strongly implicated in the pathogenesis of the disease as acetylcholine (ACh) neurotransmission governs a wide range of cognitive functions, including perception, gnosis, learning, memory, attention, orientation, and language skills. Cholinergic deficiency in AD patients has been shown to contribute to the development of cognitive and behavioral impairment [14,15]. The activity of the enzymes choline acetyltransferase (synthetic) and acetylcholinesterase (AChE) (catabolic) is significantly altered in the cerebral cortex, hippocampus, and amygdala [16]. Furthermore, several studies have demonstrated the in vivo anticholinergic effects of Aβ peptides administered directly into various brain structures without apparent immediate neurotoxicity [14,17,18,19]. Accordingly, conventional pharmacological approaches aimed at restoring normal ACh levels and cholinergic transmission in the brain have been well exploited and enforced using some of the first-line drugs for the symptomatic treatment of the disease, including the AChE inhibitors galantamine (GAL), rivastigmine and donepezil (DNPZ) [14].

Accumulating evidence over the last decade suggests a key role of melatonin, a tryptophan metabolite, and its levels as both a causative factor and a relevant biomarker of AD progression [20,21]. This pineal neurohormone has been shown to play a regulatory role in a variety of physiological processes, including aligning circadian rhythms, enhancing immune responses, and ameliorating age-related cognitive decline by exerting antioxidant, anti-inflammatory, and other neuroprotective effects [20,22,23,24]. In addition, melatonin appears to directly intervene in the pathophysiological processes of AD by inhibiting γ-secretase activity in the amyloidogenic processing pathway, as well as tau hyperphosphorylation and aggregation in the common PKC and GSK-3 kinase signaling cascades [20,25,26]. Aβ-induced neurotoxicity is additionally attenuated by accelerated protein clearance and degradation pathways, via both receptor-mediated and receptor-independent cellular mechanisms [26]. Melatonin levels and MT1 receptor density in the hypothalamic suprachiasmatic nucleus (SCN) have also been shown to decline with age in both clinical and preclinical settings, causing circadian disruption and neuropathological changes in the elderly and AD patients [21,26,27]. In light of these findings, melatonin and other structurally related indole derivatives have been used as scaffold structures in the design and synthesis of novel compounds to target etiological factors of AD [25,28,29,30].

In our previous study [31], we designed, synthesized, and evaluated 30 new compounds from two novel series of hybrid molecules based on melatonin and DNPZ, incorporating hydrazone or sulfonyl hydrazone fragments. Some of these compounds act as multifunctional ligands targeting different AD-related mechanisms. These compounds exhibited AChE inhibitory activity, antioxidant properties, minimal cytotoxicity, and good blood–brain barrier (BBB) permeability. The present study reports on the beneficial effects of the most promising analogs of the hydrazide–hydrazone series on key pathognomonic features of AD in both cellular and in vivo models. The anti-fibrillogenic properties of the tailored benzylpiperidine- (**3a**–**d**), benzylpyrrolidine- (**3e**, **3h**, **3i**), *tert*-butyl-2-hydrazinyl-2-oxoethylcarbamate derivatives (**3j**, **3k**, **3l**), indole- (**3k**, **3n**), and vanillin- (**3r**) bearing structures were evaluated using an enzyme-linked immunosorbent assay (ELISA)-based immunological analysis to identify quantitative changes in the formation of neurotoxic Aβ42 species in human neuronal cells in response to treatment. Acute toxicity studies were performed in mice after intraperitoneal (i.p.) and oral (p.o.) administration of the lead compounds **3a** and **3c**, endowed with pleiotropic anticholinesterase, antioxidant, and anti-β-amyloid activities. All experiments were carried out in a comparative manner to the reference drugs DNPZ and/or GAL, with the aim of establishing any pharmacological superiority of the multifaceted approach in battling hallmark features of neurodegeneration.

## 2. Results

### 2.1. Results from the ELISA Assay

The tested compounds from two series, hydrazide–hydrazones **3a**–**r** and sulfonyl hydrazones **6a** and **6k**, which were synthesized in our previous work [31], are shown in Figure 1.

As part of the ongoing pharmacological characterization of the most promising hydrazide–hydrazone analogs of the series, which were found to be devoid of cytotoxic activity in normal and malignant cell lines of various origin (IC50 > 300 μM) [31], an ELISA was performed for the detection and quantification of amyloidogenic Aβ42 in the supernatant of human neuroblastoma SH-SY5Y cells. Quantitative changes in the formation of neurotoxic Aβ42 species were monitored in neuronal cell cultures treated for 24 h. Biochemical analysis was performed against the untreated control and in comparison with the cholinergic drugs DNPZ and GAL (with no particular effect on Aβ fibrillogenesis) at a treatment concentration of 100 μM for all compounds.

The levels of the human amyloidogenic Aβ42 in neuronal SH-SY5Y cells after 24 h exposure to subtoxic concentrations (100 μM) of the experimental compounds and the reference drugs DNPZ and GAL relative to untreated controls (Ko) are presented in Figure 2.

The results of the highly specific ELISA immunoassay showed different quantitative changes in Aβ42 production in neuronal SH-SY5Y cell cultures after 24 h exposure to equimolar concentrations of the experimental and reference compounds (Figure 2). Different trends in the neuroprotective potential of the tested chemical analogs of the series were observed, which were probably determined by the alternating benzylpiperidine (**3a**–**d**), pyrrolidine (**3e**, **3h**, **3i**), tert-butyl-2-hydrazinyl-2-oxoethylcarbamate derivatives (**3j**, **3k**, **3l**), indole (**3k**, **3n**), and vanillyl (**3r**) fragments in their general hydrazide–hydrazone structure, as well as sulfonyl hydrazine fragment in **6a** and **6k**. Nevertheless, all compounds, except **3i,** significantly outperformed the reference drugs in terms of their Aβ42-lowering activity. The most pronounced downregulating effect on Aβ42 levels was found for the *tert*-*butyl*-2-hydrazinyl-2-oxoethylcarbamate derivatives **3k** and **3l**, which reduced Aβ42 levels by 15- and 30-fold, respectively, compared with the untreated control (Aβ42 concentration in the cell supernatant 4.89 and 10.77 pg/mL, respectively) and led to an almost complete depletion of the amyloidogenic factor. The benzylpiperidine analogs **3a** and **3b**, as well as the indole derivative **3n**, which induce a 6–7-fold reduction in target antigen levels, also show marked anti-Aβ activity. Hydrazide hydrazones with benzylpiperidine (**3c**), benzylpyrrolidine (**3e**, **3h**), imidazole (**3o**), *tert*-*butyl*-2-hydrazinyl-2-oxoethylcarbamate derivative (**3j**), and sulfonamide (**6a**) fragments reduced β-amyloid production by 200–300%, and the analogs **3d** (with a p-methoxy substituted aromatic core), **3i** (with an o-methoxy substituent), **6k** (sulfonamide group), and **3r** (vanillin group) exerted a moderate attenuating effect on the studied neurotoxin.

### 2.2. Results of the Acute Toxicity Study

Consequently, we continue to investigate the AChE inhibitors (**3a** and **3c**) that simultaneously interact with multiple targets and exhibit excellent antioxidant activities and very low cytotoxicity, possess good BBB permeability, and demonstrate marked anti-Aβ activity.

To gain insight into the in vivo safety profile of the lead compounds **3a** and **3c**, acute toxicity studies were conducted and the median lethal dose (LD50) was calculated for an intraperitoneal route of administration.

At 2000 mg/kg, 100% of the experimental animals died, at 1000 mg/kg, 66.7% of the animals died, and at 500 mg/kg, a 33.3% lethality was found. Administration of the low doses of 250 and 125 mg/kg did not cause lethality. Assuming that the lowest lethal dose is 500 mg/kg and the highest nonlethal dose is 250 mg/kg, LD_50_ for i.p. application of **3a** and **3c** was calculated as follows:LD_50_ = √(D_0_ × D_100_) = √(250 × 500) = 353.6 mg/kg for i.p. application.

Animals that survived the acute toxicity tests were observed once daily for up to 14 days. During this period, no changes were observed in social behavior in the cage or in responses to “external stimuli” (manipulation response, righting reflex, hand clapping response, response to noise or light fluctuations, response to pinching of toes or tail). No changes in the skin, fur, eyes, mucous membranes, the appearance of secretions, and autonomic activity (lacrimation, piloerection, changes in pupil size, or abnormal respiratory movements) were observed. No abnormalities in food and water consumption were observed.

On day 14 after acute intraperitoneal toxicity, all remaining live animals were sacrificed. Macroscopic examination at necropsy revealed no gross visible changes or lesions in the vital organs.

Table 1 shows the results of the blood count analysis of the experimental animals. The combination of DNPZ with SC resulted in a slight increase in WBC counts compared with the control and SC alone but remained within the reference range for mice. In the groups treated with DNPZ and the combination of SC with **3a**, platelet counts were increased compared with the control group but also remained within the reference range for mice.

Although there were some differences in hematological parameters in the experimental groups compared with the control animals, the values of the parameters studied remained within the limits of the reference values for mice.

Biochemical parameters also showed slight deviations compared with controls (Table 2). Compound **3a** reduced the level of total protein by about 30% compared with the control, which is probably related to the impaired synthetic function of the liver. Liver dysfunction is also suggested by the increased activity of ASAT in almost all experimental groups compared with the control. ASAT is a less specific enzyme for liver function, it is also found in other organs such as kidneys, pancreas, heart, muscles, etc. Therefore, its slightly elevated activity compared with the reference limits cannot be interpreted as an indicator of serious liver damage. The remaining biochemical parameters were within the reference range, indicating that the newly synthesized compounds investigated, as well as DNPZ administered i.p. over a period of 14 days, were not hepatotoxic or nephrotoxic.

### 2.3. Effects of the Tested Compounds on AChE Activity, MDA Quantity, and GSH Level in Mouse Brain Homogenate

The present study investigated the effects of two newly synthesized DNPZ-based hybrid molecules with hydrazine fragments (**3a** and **3c**) on AChE activity and the oxidative stress biomarkers malondialdehyde (MDA) and reduced glutathione (GSH) in mouse brain homogenate. The effect of the two compounds was compared with the positive control DNPZ on SC-induced brain toxicity.

Scopolamine administered alone was found to increase brain AChE activity by 44% compared with the control group (Figure 3). DNPZ, **3a**, and **3c** administered alone did not affect the activity of this enzyme. In combination with SC, DNPZ significantly reduced AChE activity by 37%, and **3a** and **3c** reduced it by 26% and 33%, respectively, compared with the SC-treated group alone.

In the present study, compound **3c** administered alone reduced the amount of MDA in the brain statistically significantly by 21.4%, while SC increased it by 76% compared with the control group (Figure 4). In animals treated with both SC and the test compounds, **3a** and **3c**, there was a statistically significant decrease in the MDA level by 19.3% and 23%, respectively, compared with the group treated with SC alone.

In terms of reduced GSH levels in the brains of the experimental animals, DNPZ and 3c administered alone increased GSH levels by 22% and 37%, respectively, compared with the control group (Figure 5). Scopolamine (3 mg/kg), administered once daily for 14 days, caused a profound 27% decrease in brain GSH levels in mice. As shown in Figure 5, the GSH depleting effect on SC was significantly inhibited by DNPZ, **3a,** and **3c** by 33%, 37%, and 72% respectively, suggesting a free radical scavenging property.

### 2.4. Effects of the Compounds Tested on Scopolamine-Induced Neuronal Damage in the Hippocampus of Mouse Brain Homogenate

Hematoxylin and eosin (H&E)-stained sections from the dorsal hippocampus of the control groups (Control, **3c**, **3a**, DNPZ) and SC-treated groups (SC, **3c** + SC, **3a** + SC, DNPZ + SCOP) are shown in Figure 6A–H.

Administration of SC to the vehicle-treated mice resulted in neuronal damage in the CA1 region (*p* = 0.032 vs. Control) and CA2 region (*p* = 0.013 vs. Control) (Figure 7A,B). Compound **3a** was only effective in the CA1 region (*p* = 0.006 vs. SC group), while **3c** had a protective effect in both the CA1 and the CA2 regions (*p* < 0.001 vs. SC group).

Scopolamine injection induced subfield-specific neuronal loss in the CA3a and CA3c subfields (*p* < 0.001 vs. Control) but not in the CA3b subfield (*p* > 0.05 vs. Control) (Figure 7A–C). In contrast to the reference drug DNPZ, which was ineffective against the SC-induced neuronal damage, compound **3c** had a neuroprotective effect in all subfields of the CA3 region (CA3a: *p* = 0.017, CA3b and CA3c: *p* < 0.001 vs. SC group, respectively). In addition, compound **3a** was effective against neuronal loss caused by SC only in the CA3c region (*p* < 0.001 vs. SC group).

Similar to the CA3a and CA3c subfields, administration of SC to the vehicle-treated mice caused neuronal damage in the GrDG (*p* = 0.003 vs. control) and PoDG (*p* < 0.001 vs. control) (Figure 7F,G). The hybrid analog **3c** had a neuroprotective effect in SC-treated mice (*p* < 0.001 vs. SC) (Figure 7F). The SC and the two compounds **3a** and **3c** did not have effect on neurons in the PoDG (Figure 7G).

## 3. Discussion

The present study is a continuation of our recent reports focused on evaluating the potential neuroprotective effects of novel compounds, specifically designed hybrids of indole and/or DNPZ, for the treatment of AD [31,36]. Intraneuronal Aβ deposition is an important pathognomonic and histopathological feature of AD, making it a suitable marker for the assessment of neuronal function and an important therapeutic target in the development of new drugs for neurodegenerative diseases [37]. Considering that the neuronal death observed in neurodegenerative diseases such as AD is partly caused by the accumulation of neurotoxic Aβ oligomers and the formation of Aβ plaques in different brain regions, we evaluated the anti-fibrillogenic properties of benzylpiperidine (**3a**–**d**), benzylpyrrolidine (**3e**, **3h**, **3i**), *tert*-butyl-2-hydrazinyl-2-oxoethylcarbamate (**3j**, **3k**, **3l**), indole (**3k**, **3n**), and vanillin (**3r**) derivatives in human neuronal cells in response to treatment. The pharmacological rationale for selecting these compounds for further investigation was their minimal neurotoxicity (IC50 > 300 μM) against both human (SH-SY5Y) and murine (Neuro-2a) malignant neuroblastoma cell lines, as well as normal murine fibroblast cells (CCL-1), as demonstrated in our previous study [31]. The lead representatives of the series were subjected to an ELISA analysis in the present survey to quantify changes in the formation of neurotoxic Aβ42 species in human neuronal cells in response to treatment. Our results showed good efficacy of most compounds in protecting SH-SY5Y cells from exposure to preformed Aβ fibrils. All selected compounds drastically diminish the generation of the neurotoxic Aβ42 oligomers in the cell supernatant to low picomolar concentrations (4.89 to 136.14 pg/mL). Benzylpiperidine analogs **3a**, **3b**, and **3c**, as well as the indole derivative **3n**, showed remarkable anti-Aβ activity with a 6–7-fold reduction in the target antigen levels. However, the most pronounced neuroprotective effect was found for the *tert*-butyl-2-hydrazinyl-2-oxoethylcarbamate derivatives **3k** and **3l**, which reduced Aβ42 levels by 15- and 30-fold, respectively, compared with the untreated control (Aβ42 concentration in the cell supernatant 4.89 and 10.77 pg/mL, respectively).

It is known that truncated Aβ42 peptides accumulate as insoluble aggregates in the AD brain because of their high resistance to degradation in the lysosomes where the lower pH environment promotes their concentration, fibrillar assembly, and deposition in a variety of cells [38]. The *N*-benzylpiperidine derivative with the acetylindole fragment 3a showed the best inhibitory activity among the compounds in this family (**3a**–**d**), followed by **3b** and **3c**. It is known from our previous work that the introduction of a methylene group (CH_2_) in the connecting hydrazone fragment in compound **3a**, compared with the indole–donepezil hybrid **3c** with the same moieties, led to a decrease in the inhibitory activity of **3a** toward AChE and an increase in activity toward the butyrylcholinesterase (BuChE) isoenzyme (the calculated selective index is in favor of the BChE (SI BChE = 1.35) [31]. Concomitantly, **3a** reduced Aβ42 levels more efficiently than **3c**. The N-benzylpiperidine containing acethyl-indole hybrid **3c** is a molecule with a reduced hydrophobic energy because it has a less hydrophobic surface, resulting from the absence of one CH_2_ group. As demonstrated by other colleagues, compounds with higher hydrophobicity can better hinder Aβ fibril assembly by interfering with key hydrophobic interactions [39]. Albeit a minor structural difference, a single CH_2_ group can account for a slight increase in the hydrophobic properties of the compound and its inhibitory capacity on Aβ aggregation. While ambiguities still remain with respect to the intrinsic physicochemical and mechanistic features of the studied molecules, the observed prominent anti-fibrillogenic activity of 3a may be attributed to its higher lipophilicity [40].

The presence of an indole moiety in compounds **3a**, **3b**, and **3c** resembling melatonin’s structure indicates that their effects may arise from their antioxidant properties as well as the activation of MT1 and/or MT2 receptors in the brain, similarly to MLT. The latter is known to exert anti-fibrillogenic effects and preserve cells from Aβ-mediated toxicity via receptor-dependent and independent pathways [31,36]. Since most of the bioactive indole–donepezil-carrying compounds **3a**–**c** have been shown to scavenge reactive oxygen species and prevent lipid peroxidation [31], their shielding effect against Aβ neurotoxicity may also be linked to the inhibition of oxidative stress [41]. Notably, the most significant reduction in the linoleic acid oxidation rate by the FTC method was demonstrated by **3a**, followed by **3c** [31]. On the other hand, compounds **3a**, **3b**, and **3c** are moderate bases like DNPZ due to the presence of a *N*-benzylpiperidine fragment and can, therefore, exert additional neuroprotective effects in AD pathology by similar to the drug mechanisms.

These findings are an encouraging step forward in the development of hybrid compounds with multimodal mechanisms of neuroprotection, prompting the need for further research of the representatives **3a**–**c** given their superior anti-amyloid activity compared with the reference drugs DNPZ and GAL (Figure 2). Conversely, pyrrolidine-containing compounds (**3e**, **3h**, **3i**) showed weaker inhibitory capacity on Aβ42 aggregation than the piperidine derivatives **3a**–**d**, possibly due to the lower basicity of the pyrrolidine ring as opposed to the piperidine system. Surprisingly, the introduction of a *tert*-butyl-2-hydrazinyl-2-oxoethylcarbamate substituent in the melatonin scaffold by a hydrazone linkage in compounds **3k** and **3l** favored a substantial decrease in Aβ42 production in the amyloidogenic pathway although these analogs have not demonstrated sufficient acetyl- and butyrylcholinesterase inhibitory activity in our previous study [31]. The observed anti-amyloidogenic activity of these compounds **3k** and **3l** carrying a melatonin fragment (Figure 1 and Figure 2) suggests the recruitment of alternative neuroprotective pathways that must be added to their excellent antioxidant profile, as reported for **3k** (outperforming BHT in DPPH in all test methods) [31]. Presumably, the *tert*-butyl-2-hydrazinyl-2-oxoethylcarbamyl group is of detrimental importance in diverting Aβ42 production and self-induced aggregation [42].

As for the other tested derivatives (**3n**, **3o**, **3r** and **6a**, **6k**), the results from the quantitative β-amyloid study are indicative of several other possible structure–activity relationships, according to which the presence of methoxy- and hydroxy- groups in the aromatic rings of **3n**, **3o**, **6k**, and a vanillyl core in **3r** are irrelevant to the occurrence of neuroprotection. Thus, according to the general structure–activity profile of the compounds, the introduction of two indole units in the derivatization of **3n** and of methoxy- and hydroxy-substituted benzene rings in **3o** and **3r** lessened their Aβ42 inhibitory activity compared with **3k** and **3l**. The presence of a sulfonyl hydrazone scaffold between the aromatic fragments in **6a** and **6k** was also unfavorable to their neuroprotective potential (Figure 2). However, the 24 h exposure of the neuronal SH-SY5Y cells to subtoxic concentrations of the lead compounds **3k** and **3l** as well **3a**–**c** was able to induce a drastic downregulation of Aβ42 levels to the order of tenfold. Based on these findings, the most promising hybrids **3a** and **3c** were investigated in the mouse scopolamine-induced dementia model to further evaluate their multimodal Aβ42-reducing properties, as well as excellent antioxidant and acetylcholinesterase inhibitory activities at the in vivo level.

While DNPZ has been shown to improve cognitive function, it has failed to produce a positive effect on several biomarkers associated with AD and insulin resistance in type 2 diabetic (T2D) rats, as reported by [43] who found that the beneficial neuroprotective effect of DNPZ in brain regions vulnerable to epilepsy is dependent on the treatment schedule. However, DNPZ exerted promising neuroprotective effects against brain injury and AD pathology induced by cardiac ischemia/reperfusion injury, regardless of the timing of treatment initiation. These findings highlight the need for further research to develop more effective treatment strategies.

According to the Hodge and Sterner scale [44], the compounds tested can be classified as mildly toxic (500–5000 mg/kg body weight) when administered orally to male mice and moderately toxic (50–500 mg/kg) when administered i.p. These LD50 values may also be related to the mechanism of action of the compounds, namely, inhibition of AChE. It can be speculated that their low oral toxicity may be partly due to poor oral bioavailability owing to their pharmacokinetic properties. First, it is possible that the tested substances are not readily absorbed in the gastrointestinal tract and are excreted in the feces. Second, there might be extensive hepatic and/or intestinal biotransformation involved where the compounds are metabolized to nontoxic metabolites. To evaluate these possibilities, more specific pharmacokinetic studies are required to determine the plasma concentrations of metabolites and parent compounds and to measure their concentrations in urine and feces. The calculated biochemical parameters for **3a** and **3c** showed minor deviations compared with the reference drug DNPZ, such as the total protein level and ASAT activity, which suggest that they are not associated with hepatotoxic or nephrotoxic activity.

Oxidative stress is an important pathophysiological mechanism contributing to the neurodegeneration of central cholinergic pathways in AD. It has been established that reduced brain ACh concentration is not the primary pathophysiological cause of AD but rather a consequence of the disease progression. Dysfunctional cholinergic neurotransmission and reduced brain ACh levels are associated with the accumulation of Aβ, which mediates neuronal inflammation. There is scientific evidence that SC-induced brain toxicity and dementia in rodents are also associated with oxidative stress and neuroinflammation [45]. AChE inhibitors such as DNPZ, GAL and other drugs that increase central cholinergic neurotransmission are used in therapeutic regimens for the symptomatic treatment of the disease. In addition to this mechanism, there is evidence that AChE inhibitors exert neuroprotective effects by reducing oxidative stress and free radical-induced inflammation in the brain [45,46]. In our previous study [31], we first evaluated the in vitro antioxidant properties and AChE inhibitory activity of two series of DNPZ-based hybrid molecules with hydrazone (**3a**–**r**) or sulfonylhydrazone (**6a**–**l**). Among these compounds, two compounds, **3a** and **3c**, showed a well-balanced multifunctional pharmacodynamic profile with promising antioxidant and AChE inhibitory activity. The observed antioxidant capacities of these derivatives may be due to their ability to act as proton donors, thereby neutralizing free radicals and/or inhibiting lipid peroxidation (LP). LP is the most important type of oxidative radical damage in biological systems due to its interplay with ferroptosis and its role in the secondary damage of other biomolecules such as proteins [47]. Oxidative stress induces neuronal damage through LP, DNA damage, protein oxidation, and inflammation. SC is an antimuscarinic drug that works by blocking ACh in the central nervous system (CNS) [48]. It impairs learning and memory by increasing the activities of AChE, BuChE, and adenosine deaminase (ADA) and promoting LP, while reducing nitric oxide (NO) and reduced GSH levels, superoxide dismutase (SOD), glutathione S-transferase (GST) and catalase activities. DNPZ is known to inhibit the activity of BuChE and ADA, reduce LP and increase NO levels, and enhance antioxidant status. It also regulates the endocytic trafficking of APP by upregulating the expression of sorting nexin protein-33 (SNX33) [49,50]. The present study showed that the two lead compounds, and in particular **3c**, have AChE inhibitory and antioxidant activity comparable to the reference drug DNPZ in the mouse SC model. These results confirm in vitro data from previous studies in which Angelova et al. [31] found that compound **3c** exerts a potent inhibitory effect on the enzyme that degrades ACh. In addition, both compounds have recently shown very good BBB permeability and promising antioxidant activity as assessed by chemical methods.

In contrast to DNPZ, both compounds, **3a** and **3c**, were able to protect cells in the dorsal hippocampus against SC-induced neurotoxicity (Figure 6 and Figure 7). Furthermore, the **3c** hybrid showed superior neuroprotection to the **3a** compound in all subfields except the PoDG, whereas the **3a** compound showed beneficial effects against SC neurotoxicity only in the CA1 and CA3c regions. Notably, the reference drug DNPZ failed to deliver any neuroprotective effects against SC-induced toxicity in the regions of the dorsal hippocampus examined. Previous reports have shown controversial data regarding the neuroprotective activity of this AChE inhibitor tested in various rodent models of neurodegeneration [43,51,52].

## 4. Materials and Methods

### 4.1. ELISA Assay

A “sandwich” ELISA immunoassay was conducted to monitor changes in Aβ42 levels in the supernatant of human neuroblastoma SH-SY5Y cells according to the manufacturer’s instructions (Human Amyloid Beta 42/AB 1-42, Assay Genie^®^, Dublin, Ireland). The quantitative changes in the formation of neurotoxic Aβ42 species were monitored in neuronal cell cultures treated for 24 h with 15 of the most promising hydrazide–hydrazone analogs of the series that were found to be devoid of cytotoxic activity in normal and malignant cell lines of different origin (IC50 > 300 μM) [31]. Biochemical analysis was performed against untreated control and in comparison with the cholinergic drugs DNPZ and GAL at a treatment concentration of 100 μM for all compounds.

The method is highly specific and sensitive and widely used in immunology and biochemistry to quantify the concentration of a specific antigen in a sample using antibodies. A specific capture antibody against Aβ42 is immobilized to a solid-phase carrier (microtiter plate) into the wells of which standards, controls, and cell supernatants from treated samples were loaded, respectively. After the indicated incubation period and washing procedures, a specific biotinylated detection antibody binding Aβ42 to a different epitope was added to the wells. Following re-incubation and washing, a streptavidin-peroxidase conjugate (Avidin-HRP) was added, which interacted with a chromogenic substrate tetramethylbenzidine (TMB) and served as an indicator system to visualize the streptavidin-peroxidase reaction. A blue coloration was obtained, which changed to yellow upon the addition of an acid-stop solution. The concentration of Aβ42 was directly proportional to the intensity of the color. Absorbance was measured spectrophotometrically using an ELISA Reader at 450 nm. The antigen concentration in the test samples was calculated by reference to a standard calibration curve constructed based on the absorbance signals of standard serial dilutions of the antigen within the concentration range of 0–1000 pg/mL.

#### 4.1.1. Cell Lines and Culture Conditions

Human neuronal cell line SH-SY5Y, widely used as an in vitro model to study neuronal function, differentiation, and neurodegenerative processes, was used as an in vitro model in measuring the anti-β-amyloid activity of the experimental compounds. Cells were purchased from the tumor cell bank DSMZ—Deutsche Sammlung von Mikroorganismen und Zellkulturen GmbH (German Collection of Microorganisms and Cell Cultures), Braunschweig, Germany. Cell cultures were grown under standard conditions in an incubator (Heraeus BB 16-Function Line, Budapest, Hungary) at 37 °C, humidified medium and 5% carbon dioxide and maintained in the appropriate culture medium specified by the provider.

#### 4.1.2. Compounds

Fifteen new compounds (**3a**, **3b**, **3c**, **3d**, **3e**, **3i**, **3h**, **6k**, **6a**, **3o**, **3r**, **3n**, **3l**, **3k**, **3j**) structurally characterized in a previous study [31], were evaluated for their neuroprotective properties. All solvents, chemicals, and reagents were obtained commercially and used without purification. The reference drugs DNPZ and GAL were purchased from Sigma-Aldrich (St. Louis, MO, USA).

### 4.2. In Vivo Acute Toxicity Study

#### 4.2.1. Experimental Animals

The experiments used 78 male ICR mice (6 weeks old, 25–30 g) obtained from the Experimental and Breeding Base for Experimental Animals, part of the Institute of Neurobiology, Bulgarian Academy of Sciences. The mice were kept under standard laboratory conditions (ambient temperature 20 °C ± 2 °C and relative humidity 72% ± 4%, 12 h light/dark cycle). All procedures involving the animals were pre-approved by the local Animal Welfare Committee and the Bulgarian Food Safety Authority (Authorization No. 347 and Opinion No. 263) in accordance with the European Community Guidelines for the Use of Animals in Laboratories (Directive 2010/63/EU). All the experimental procedures were carried out by licensed specialists at the Faculty of Veterinary Sciences of the Sofia Forestry University. Efforts were made to minimize animal suffering.

#### 4.2.2. Drugs and Chemicals

Donepezil (DNPZ) (Pfizer, Manhattan, NY, USA) and scopolamine hydrobromide trihydrate 96% (SC) (Across organics, Shanghai, China), were used in the study. Doses of both pharmaceutical compounds used in this study were chosen based on the information obtained from previous investigations [53,54].

#### 4.2.3. Design of the In Vivo Experiment

The acute toxicity of the studied compounds was evaluated in 30 male mice after i.p. administration of compounds **3a** and **3c** according to Lorke’s method [55]. Five fixed dose intervals (2000, 1000, 500, 250, and 125 mg/kg) and three animals per dose were used. Doses were selected according to published predictive in silico LD_50_ [31]. For better solubility, compounds were solubilized with Tween 80 (0.1%) and then further sonicated.

LD_50_ was calculated using the equation,
LD_50_ = √(D_0_ × D_100_) = √(250 × 500) = 353.6 mg/kg for i.p. application. 
where D_0_ is the highest nonlethal dose and D_100_ is the lowest lethal dose.

Surviving animals were observed for 24 h and then for up to 14 days. On day 14, the remaining live animals were euthanized after anesthesia with ketamine/xylazine (80/10 mg/kg, i.p.) and internal organs were inspected for possible macroscopic abnormalities (organ color, consistency, neoplasms, etc.).

Scopolamine brain toxicity was induced in male ICR mice by i.p. injection of 3 mg/kg SC for 14 days. Tested compounds were also administered i.p. for 14 days at doses representing 1/10 of the LD_50_ or 35 mg/kg. It would be expected that this route of administration will improve the bioavailability of the study compounds in the brain by jumping the GIT barrier or GIT metabolism.

To assess the effects of test compounds on SC-induced changes in brain biochemical parameters, 48 male mice were divided into eight groups, with 6 mice in each group (n = 6).

**Group 1**—control animals, treated with vehicle (0.9% saline i.p.) for 14 days;**Group 2**—animals treated with positive control DNPZ (1 mg/kg i.p.) for 14 days;**Group 3**—animals treated with the tested compound **3a** alone at a dose 1/10 LD_50_ or 35 mg/kg i.p. for 14 days;**Group 4**—animals treated with the tested compound **3c** alone at a dose 1/10 LD_50_ or 35 mg/kg i.p. for 14 days;**Group 5**—animals treated with SC alone (3 mg/kg i.p.) for 14 days;**Group 6**—animals treated with SC + DNPZ combination for 14 days;**Group 7**—animals treated with SC + **3a** combination for 14 days;**Group 8**—animals treated with the combination of SC + **3c** for 14 days.

#### 4.2.4. Measurement of Acetylcholinesterase (AChE) Inhibition in Brain Homogenate

Brains were homogenized with 0.1 M phosphate buffer, pH 7.4. Aliquots of brain homogenates from different groups were used to measure AChE activity for 10 min by the method of Ellman [56]. AChE activity was calculated and expressed as nmol/min/mg protein using a molar absorption coefficient of 13,600 M^−1^ cm^−1^.

#### 4.2.5. Measurement of Malondialdehyde (MDA) Levels in Brain Homogenate

Brains were homogenized with 0.1 M phosphate buffer and EDTA, pH 7.4. Aliquots of the homogenates were heated for 20 min on a water bath (100 °C) with thiobarbituric acid. The amount of thiobarbituric acid-formed reactive species (TBARS) (expressed as MDA equivalents) was measured spectrophotometrically by the method of Deby and Goutier [57] at a wavelength of 535 nm. The concentration of MDA was calculated using a molar absorption coefficient of 1.56 × 10^5^ M^−1^ cm^−1^ and expressed in nmol/g tissue.

#### 4.2.6. Measurement of Glutathione (GSH) Levels in Brain Homogenate

GSH was evaluated by measuring nonprotein sulfhydryls after trichloroacetic acid (TCA) protein precipitation by the method described by Bump et al. [58]. Brains were homogenized in 5% TCA (1:10) and centrifuged for 20 min at 4000× *g*. The reaction mixture contained 0.05 mL supernatant, 3 mL 0.05 M phosphate buffer (pH = 8), and 0.02 mL DTNB reagent. Absorption was determined at a wavelength of 412 nm and the results were expressed as nmol/g tissue.

#### 4.2.7. Measurement of Hematological and Serum Biochemical Data

Whole blood was analyzed by a semi-automated hematological analyzer BC-2800 Vet, (Mindray, Shenzhen, China) according to the manufacturer’s instructions. The count of leukocytes (WBC), erythrocytes (RBC, Er), platelets (PLT), amount of hemoglobin (Hb), and hematocrit (Ht) were measured. The biochemical serum data as blood sugar (GLU), urea (UREA), creatinine (CREAT), total protein (TP), albumin (ALB), uric acid (Uric acid), as well as the activity of the enzymes amylase (AMYL), aspartate aminotransferase (ASAT), and alanine aminotransferase (ALAT), were measured using automated biochemistry analyzer kits (BS-120, Mindray, Shenzhen, China), following the manufacturer’s instructions.

#### 4.2.8. Pathomorphological Evaluation of Brain Tissue Specimens

Tissues from the brains of the mice from all groups were collected post mortem and fixed in 10% buffered formalin for 48 h. After post-fixation, the tissue blocks were washed in distilled water, dehydrated in an ascending series of alcohols, embedded in paraffin, cut into 5 μm thick coronal sections, and mounted on chrome-gelatin-coated slides. The tissue sections were then deparaffinized with xylene and ethanol and routinely stained with hematoxylin and eosin to better identify pyknotic nuclei of damaged neurons. After staining, the sections were dehydrated in ethanols, cleared in xylene, and embedded in Entellan.

#### 4.2.9. Photodocumentation and Image Analysis

After classical hematoxylin and eosin staining, the slides were scanned at magnification 20× with an Aperio AT2 linear scanning microscope (Leica Biosystems, Wetzlar, Germany) with software Aperio ImageScope (Aperio ImageScope software version 12.4.6.5003, Leica Biosystems GmbH, Nussloch, Germany) during image acquisition. To quantify cells, the stained sections were digitalized to produce scans of the respective sections and regions of interest (ROIs). The scanned images were analyzed using the program ViewPoint Light (version 1.0.0.9628, PreciPoint, Freising, Germany). The area of the interest was outlined manually, and the cells of interest were counted by an experienced researcher blinded to the treatment groups.

#### 4.2.10. Statistical Analysis

The statistical program ‘MEDCALC’ was used for the analysis of the in vivo data. The results are expressed as mean ± SD for six animals in each group. The significance of the data was assessed by the Bonferroni-corrected Mann–Whitney *U* test. Values of *p* < 0.01 were considered statistically significant.

## 5. Conclusions

The present research provides a thorough theoretical and practical evaluation of multi-targeted indole/donepezil hybrids with tailored structures and neuroprotective properties against key pathogenetic factors in AD. A detailed pharmacological characterization of the tested compounds was conducted both in vitro and in vivo and revealed important structure–activity relationships accountable for their complex mechanistic behavior. As lead compounds among the tested series emerged **3a** and **3c**, which showed remarkable anti-amyloidogenic, antioxidant and AChE inhibitory activities. Furthermore, both compounds effectively influenced brain injury and AD-like pathology in the SC-induced dementia model in mice by attenuating cholinergic damage, LP, and neuronal loss in the hippocampus. These results are consistent with the recently reported findings of our team that the compound 3c showed promising anti-pathogenic activity in a rat model of pinealectomy + icvAβ1-42 on AD biomarkers Aβ1-42 and pTAU in the hippocampus that was comparable to the effects of the reference drug melatonin [36]. Furthermore, in this in vivo rat model of AD we have found that **3c** supplementation facilitated non-amyloidogenic signaling via receptor-related signaling (MT1A and MT2B/ERK/CREB). The data obtained from this study may serve as a proof of concept for the therapeutic potential of the multifaceted approach in molecular design, aimed at integrating synergistic beneficial properties of several pharmacophore groups in a single hybrid molecule.

## Figures and Tables

**Figure 1 ijms-25-05969-f001:**
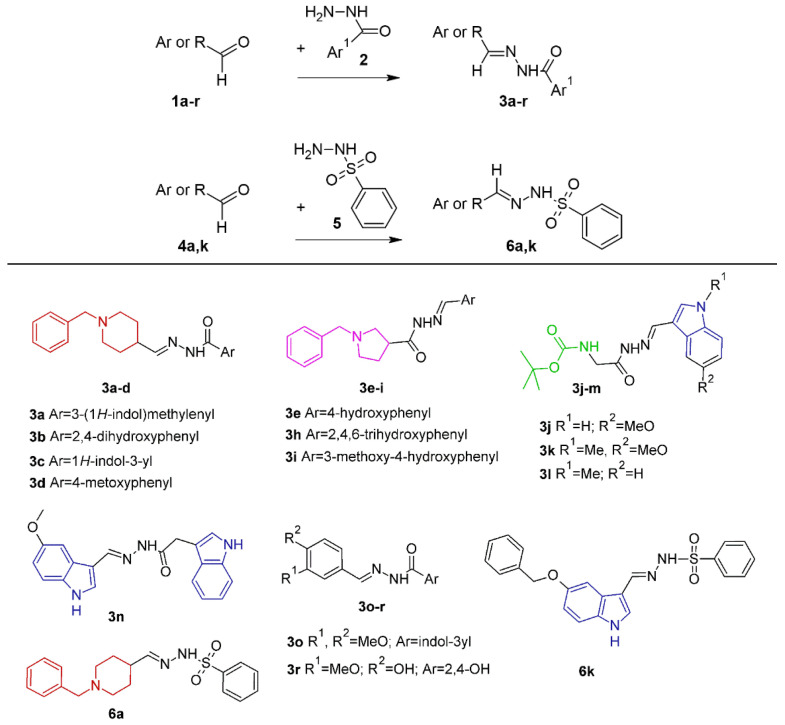
Synthetic scheme for the preparation of hydrazide-hydrazones (**3a**–**r**) and sulfonylhydrazones (**6a**–**k**) from aldehydes (**1a**–**r** or **4a,k**) and hydrazides (**2**) or sulfonylhydrazide (**5**) as described in our previous work [31] and molecular structures of the investigated hybrid hydrazide–hydrazone compounds based on benzylpiperadine (**3a**–**d**), pyrrolidine (**3e**, **3h**, **3i**), tert-butyl-2-hydrazinyl-2-oxoethylcarbamate derivatives (**3j**, **3k**, **3l**), indole (**3n**, **3o**), and vanillyl (**3r**) fragments, as well sulfonylhydrazones (whit benzylpiperadine fragment 6a and with indole fragment **6k**) [31].

**Figure 2 ijms-25-05969-f002:**
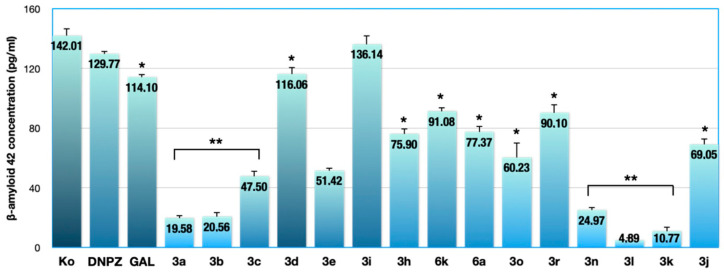
Levels of human amyloidogenic Aβ42 in neuronal SH-SY5Y cells after 24 h exposure to subtoxic concentrations (100 μM) of the experimental compounds and reference drugs donepezil (DNPZ) and galantamine (GAL) relative to untreated control samples (Ko). All samples were tested in triplicate, and statistical analysis was performed using Student’s *t*-test in GraphPad Prism (8.0) (GraphPad Software, San Diego, CA, USA). *p* values (* *p* < 0.05, ** *p* < 0.01) were considered statistically significant.

**Figure 3 ijms-25-05969-f003:**
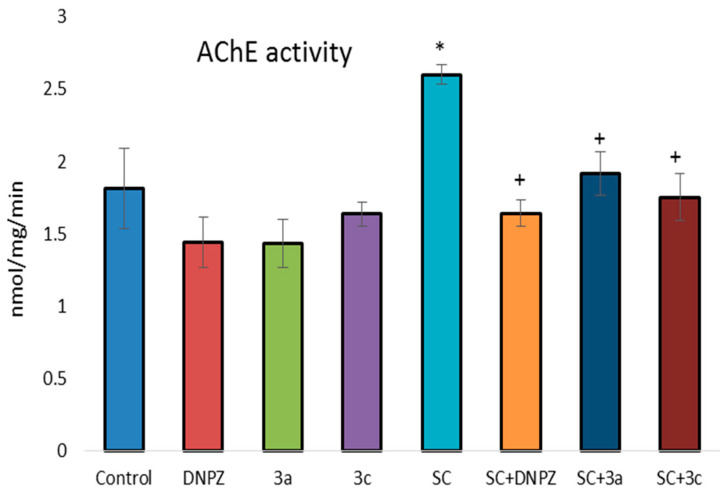
Effect of DNPZ, **3a**, and **3c**, administered alone and on the SC-induced increase of AChE activity in mice brain homogenate. * *p* < 0.01 vs. control; ^+^ *p* < 0.01 vs. SC. Results are expressed as mean ± SD (n = 6). The significance of the data was assessed using the Bonferroni-corrected Mann–Whitney *U* test. Values of *p* ≤ 0.01 were considered statistically significant.

**Figure 4 ijms-25-05969-f004:**
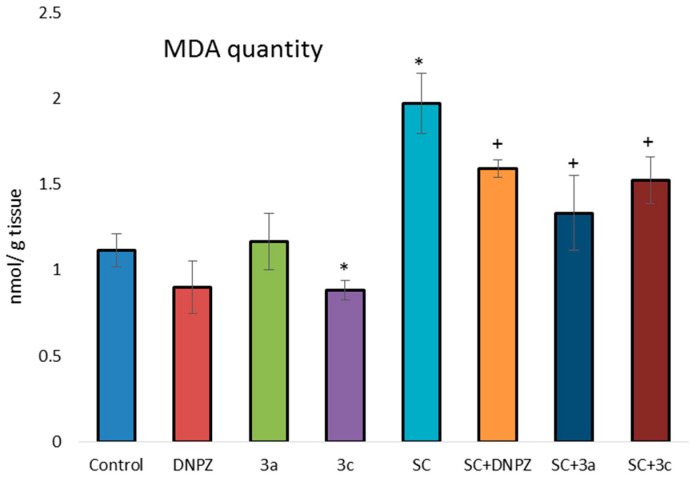
Effect of DNPZ, **3a**, and **3c**, administered alone and on the SC-induced MDA formation in mice brain homogenate. * *p* < 0.01 vs. control; ^+^ *p* < 0.01 vs. SC. Results are expressed as mean ± SD (n = 6). The significance of the data was assessed using the Bonferroni-corrected Mann–Whitney *U* test. Values of *p* ≤ 0.01 were considered statistically significant.

**Figure 5 ijms-25-05969-f005:**
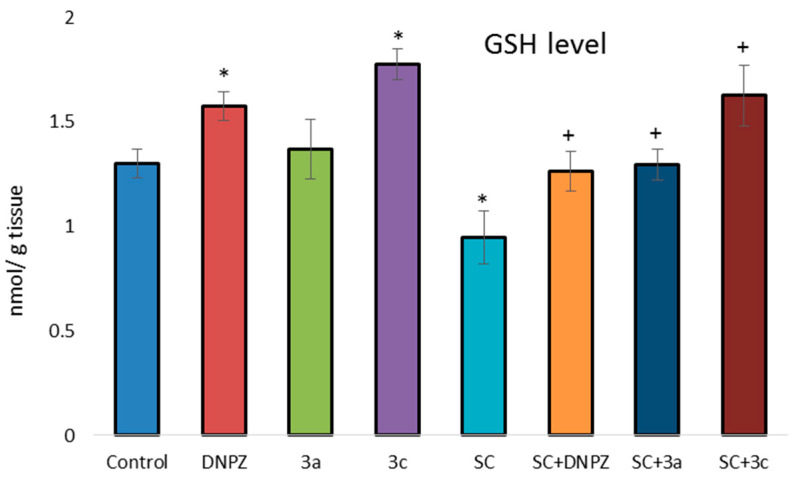
Effect of donepezil, **3a**, and **3c**, administered alone and on the SC-induced depletion of GSH in mice brain homogenate. * *p* < 0.01 vs. control; ^+^ *p* < 0.01 vs. SC. Results are expressed as mean ± SD (n = 6). The significance of the data was assessed using the Bonferroni-corrected Mann–Whitney *U* test. Values of *p* ≤ 0.01 were considered statistically significant.

**Figure 6 ijms-25-05969-f006:**
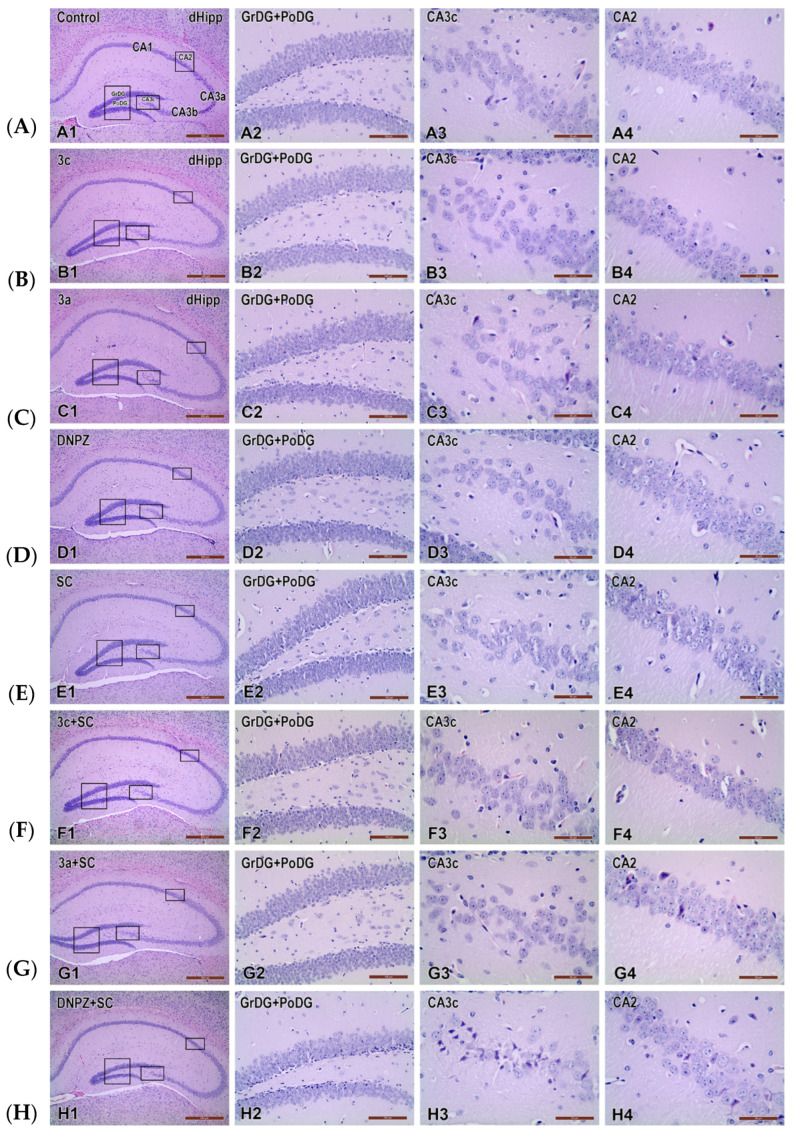
Representative photomicrographs of the dorsal hippocampal formation in control groups [Control (**A1**), **3c** (**B1**), **3a** (**C1**) and DNPZ (**D1**)] and SC-treated groups [SC (**E1**), **3c** + SC (**F1**), **3a** + SC (**G1**) and DNPZ + SC (**H1**)] stained with hematoxylin and eosin (H&E) (**H**, shown at low-power magnification in the first column of coronal sections. In the second column (**A2**–**H2**), at a high-power magnification, the granular (GrDG) and polymorphous (PoDG) layers of the dentate gyrus (DG) are presented indicated by a rectangle in the overview pictures from the first column. In the third (**A3**–**H3**) and fourth columns (**A4**–**H4**), CA3c and CA2 regions of the dorsal hippocampus are presented, respectively, of the enclosed areas from the first column. Scale bars: 500 µm (**A1**–**H1**), 100 µm (**A2**–**H2**), 50 µm (**A3**–**H3**, **A4**–**H4**).

**Figure 7 ijms-25-05969-f007:**
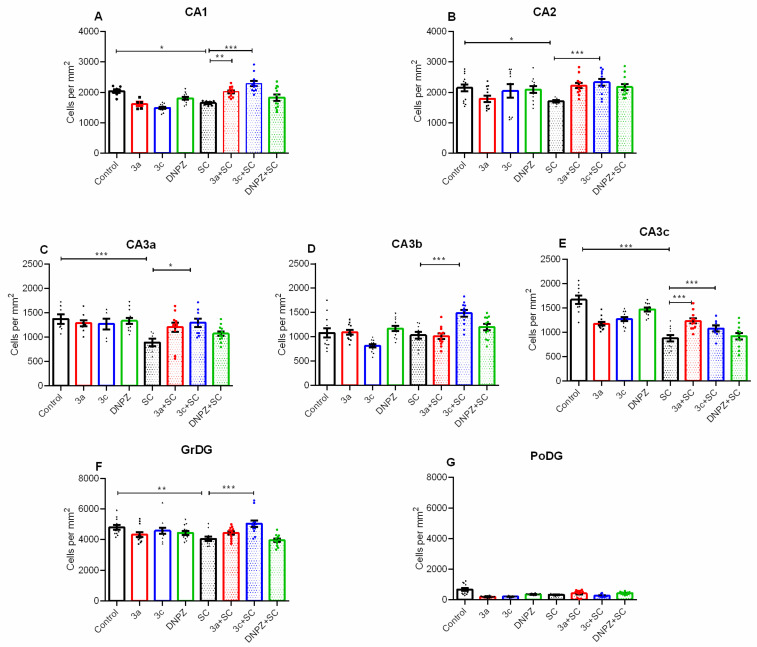
Effect of compound **3a**, **3c**, and DNPZ treatment in control and SC-treated animals on the histology scores. Neuronal damage in the CA1 (**A**), CA2 (**B**), CA3a (**C**), CA3b (**D**), CA3c (**E**) regions, granular dentate gyrus (GrDG) (**F**), polymorphic (Po)DG (**G**). * *p* = 0.032, SC vs. control; ** *p* = 0.006, **3a**-SC vs. SC group, *** *p* < 0.001, **3c**-SC vs. SC group (**A**); * *p* = 0.013, SC vs. control; *** *p* < 0.001, **3c**-SC vs. SC group (**B**); *** *p* < 0.001, SC vs. Control; * *p* = 0.017, **3c**-SC vs. SC (**C**); *** *p* < 0.001, **3c**-SC vs. SC group (**D**); *** *p* < 0.001, SC vs. Control; *** *p* < 0.001, **3a**-SC and **3c**-SC vs. SC (**E**); ** *p* = 0.003, SC vs. control; *** *p* < 0.001, **3c**-SC vs. SC (**F**).

**Table 1 ijms-25-05969-t001:** Hematological parameters in all treated groups.

Hematology	Controls	3a (35 mg/kg, i.p.)	3c (35 mg/kg, i.p.)	DNPZ (1 mg/kg, i.p.)	SC (3 mg/kg, i.p.)	SC (3 mg/kg, i.p.) + DNPZ (1 mg/kg, i.p.)	SC (3 mg/kg, i.p.) + 3a (35 mg/kg, i.p.)	SC (3 mg/kg, i.p.) + 3c (35 mg/kg, i.p.)	Ref. Range (mice)
WBC ×10^3^/µL	9.75 ± 2.3	7.65 ± 2.27	8.63 ± 1.03	8.25 ± 1.21	10.63 ± 3.46	12.03 ± 2.62 *	8.7 ± 1.42	9.12 ± 1.85	2.9–15.3
RBC ×10^6^/µL	9.11 ± 0.17	9.00 ± 0.37	8.69 ± 0.69	9.05 ± 0.17	8.98 ± 0.22	8.58 ± 0.42	8.00 ± 0.64	8.55 ± 0.37	5.6–10.4
Hgb g/L	153 ± 8.87	157 ± 6.88	154.8 ± 6.1	153.5 ± 4.8	159.5 ± 8.1	151.3 ± 6.2	133 ± 4.5	144.5 ± 6.8	120–160
HCT %	43.15 ± 2.6	44.03 ± 1.95	44.83 ± 2.95	41.1 ± 1.21	45.33 ± 3.59	41.65 ± 1.15	40.15 ± 4.94	41.63 ± 1.62	36–52
PLT ×10^3^/µL	437 ± 30.1	487.3 ± 41.2	395.8 ± 24.5	618 ± 32.2 *	426.8 ± 48.4	435.5 ± 24.5	618.5 ± 30.1 *	430 ± 37.6	127–939

* *p* < 0.01 vs. control. Results are expressed as mean ± SD (n = 6). The significance of the data was assessed using the Bonferroni-corrected Mann–Whitney *U* test. Values of *p* ≤ 0.01 were considered statistically significant. Abbreviations: DNPZ, donepezil; SC, scopolamine; SC + DNPZ: scopolamine + donepezil; SC + **3a**, SC + compound **3a**; SC + **3c**, SC + compound **3c**; WBC, white blood cells; RBC, red blood cells; Hgb, hemoglobin; HCT, hematocrit; PLT, platelets.

**Table 2 ijms-25-05969-t002:** Serum biochemical parameters in all treated groups.

Blood Biochemistry	Controls	3a (35 mg/kg, i.p.)	3c (35 mg/kg, i.p.)	DNPZ (1 mg/kg, i.p.)	SC (3 mg/kg, i.p.)	SC (3 mg/kg, i.p.) + DNPZ (1 mg/kg, i.p.)	SC (3 mg/kg, i.p.) + 3a (35 mg/kg, i.p.)	SC (3 mg/kg, i.p.) + 3c (35 mg/kg, i.p.)	Ref. Range (mice)
GLU mmol/L	8.95 ± 0.82	9.41 ± 0.77	8.95 ± 0.76	10.3 ± 0.84	10.96 ± 0.9	9.86 ± 0.79	10.33 ± 0.68	9.21 ± 0.56	4.2–11.6 [32]
UREA mmol/L	12.01 ± 0.32	9.73 ± 0.36	10.41 ± 0.28	10.5 ± 0.22	11.37 ± 0.4	8.33 ± 0.33	10.58 ± 0.45	7.59 ± 0.52	3.8–12.3 [32]
CREAT µmol/L	34.9 ± 2.3	31.8 ± 28.8	28.6 ± 6.6	32.6 ± 5.6	34.4 ± 4.8	33.6 ± 3.2	31.4 ± 3.6	41.5 ± 3.5	35–53 [32]
TP g/L	58.1 ± 2.2	40.7 ± 3.1	59.6 ± 2.6	59.1 ± 3.6	58.5 ± 3.8	61 ± 5.6	60.2 ± 6.3	60.8 ± 5.7	53–63 [32]
ALB g/L	35.8 ± 1.8	35.2 ± 1.7	33.4 ± 2.2	35.8 ± 3.1	37.6 ± 2.2	36.6 ± 2.8	36.2 ± 3.2	34.8 ± 2.6	26–39 [33]
ASAT U/L	313 ± 4.5	383.7 ± 5.2 *^+^	382.3 ± 3.6 *^+^	321 ± 4.1	320.5 ± 6.8	368.8 ± 7.2 *^+^	365.7 ± 6.3 *^+^	369 ± 5.8 *^+^	57–329 [33]
ALAT U/L	103.8 ± 2.2	70.5 ± 3.1	116.8 ± 3.3	89.2 ± 3.4	107 ± 4.5	87 ± 6.3	83.1 ± 7.2	94.8 ± 7.7	7–227 [34]
AMYL U/L	3363 ± 42.3	1464 ± 38.3	1224 ± 48.3	1909 ± 28	1386 ± 22	1321 ± 42.3	1501 ± 52.1	2155 ± 55.2	1512–3084 [34]
Uric acid µmol/L	83.6 ± 4.4	72.3 ± 2.3	177 ± 3.7 *	120 ± 6.7 *	88.3 ± 3.3	119 ± 7.8 *	115 ± 8.2 *	68.2 ± 4.4	0.1–760 [35]

* *p* < 0.01 vs. control; ^+^ *p* < 0.01 vs. reference range. Results are expressed as mean ± SD (n = 6). The significance of the data was assessed using the Bonferroni-corrected Mann–Whitney *U* test. Values of *p* ≤ 0.01 were considered statistically significant. Abbreviations: GLU, glucose; CREAT, creatinine; TP, total protein; ALB, albumin; ASAT, aspartate aminotransferase; ALAT, alanine aminotransferase; AMYL, amylase.

## Data Availability

Data is contained within the article.

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
