# Peer review of "Tailored Melatonin- and Donepezil-Based Hybrids Targeting Pathognomonic Changes in Alzheimer’s Disease: An In Vitro and In Vivo Investigation"

_ijms, 2024, doi:10.3390/ijms25115969_

Round 1

Reviewer 1 Report

Comments and Suggestions for Authors

The manuscript focuses on the further characterization, mainly in vivo or ex vivo, of two derivatives that were selected among a more extensive series of derivatives whose design, synthesis, and in vitro evaluation were previously published: Angelova, V. T. et al. Design, Synthesis, In Silico Studies and In Vitro Evaluation of New Indole- and/or Donepezil-like Hybrids as Multitarget-Directed Agents for Alzheimer’s Disease. Pharmaceuticals 2023, 16, 1194

Hence, to sum up, this manuscript is essentially a follow-up of the above-mentioned medchem work, and it is restricted to the deeper evaluation of 2 derivatives, namely 3a and 3c, which have been selected on the basis of properties reported in Pharmaceuticals 2023, 16, 1194. The reported effect on amyloid levels is a piece of work that hardly fits here and should have been included in the previously published manuscript. Indeed, the selection of 3a and 3c for in vivo studies is not based (exclusively) on the outcomes reported in Table 2, in which compounds 3l and 3k are the most promising ones.

It seems like the authors would like to fit all the data they have collected and were not able to include in the manuscript published in 2023, but this makes the incipit of the result and of the discussion sections in this manuscript not linear by reporting results on 15 compounds that do not have any follow-up and for which we miss their in vitro profile. This also forces the authors to continuously refer to previously published data and discussions on previously published data. Hence, my suggestion is to rearrange the manuscript by only focusing on the 3a and 3c derivatives, without any discussion on the amyloid-lowering properties of compounds that have not been subjected to further investigations in this manuscript. The main text should show any data on amyloid levels of the two compounds, while all the others should be removed because they have no impact on the selection. Introduction, results, and discussion sections should be rearranged consequently.

In tables reporting results from all in vivo or ex vivo experiments, it is important to indicate the administered doses of the reference compound, namely donepezil, galantamine, and 3a and 3c. Indeed, it seems that not the same administration dose was used (1mg/kg vs. 35 mg/kg). If this information is not specified and not clearly highlighted in the test, the reader gets a false perception of the relative properties of the two selected derivatives. Doses administered should also be clearly motivated and discussed.

Conclusions should be consistent with the data, as discussed above, and clearly highlight the study's limitations in light of the two following points: relative doses administered and the absence of in vivo proof with AD models of anti-amyloidogenic activity.

Further comments:

The authors should add statistical analysis (e.g., an ANOVA test) to Figure 2 and add the number of repeats to the figure label. Furthermore, the acronym identification of a given compound should be kept consistent. In the Figure 2 label, donepezil is acronymized as DNPZ, but in the bar graph, it is labeled as D. Please be uniform.

Concerning the selection of donepezil and galantamine as reference standards for the anti-amyloidogenic activity reported in Figure 2, it must be underlined that those compounds are not known as anti-aggregating agents but anticholinesterase ones.

Also, attention must be paid to the claim of neuroprotective action, which should be limited to experiments in which neuroprotection has been effectively evaluated. In detail, regarding the current discussion of Figure 2, the authors should correct the sentence: "The most pronounced neuroprotective effect was found for the tert-butyl-2-hydrazinyl-2-oxoethylcarbamate derivatives 3k and 3l..." Indeed, Figure 2 does not refer to the neuroprotection activity (no cell viability was assayed) of the tested compounds but to their ability to inhibit the amount of amyloid in the supernatant.

Please check the correct use of capital letters in the AChE acronym.

Experimental part: It is important to make readers able to repeat experiments. Hence, all applied procedures are described in detail, specifying volumes, times, buffer composition, and solution concentration. They cannot be general (see section 4.1 and subsections which do not list any detail on stock solution preparation, concentrations, solvents etc)

Comments on the Quality of English Language

Check for minor corrections

Author Response

Thank you for the careful evaluation of our manuscript. We have revised the manuscript, taking into account the suggested modifications. All changes in the MS are highlighted in red.

Point #1 The manuscript focuses on the further characterization, mainly in vivo or ex vivo, of two derivatives that were selected among a more extensive series of derivatives whose design, synthesis, and in vitro evaluation were previously published: Angelova, V. T. et al. Design, Synthesis, In Silico Studies and In Vitro Evaluation of New Indole- and/or Donepezil-like Hybrids as Multitarget-Directed Agents for Alzheimer’s Disease. Pharmaceuticals 2023, 16, 1194

Hence, to sum up, this manuscript is essentially a follow-up of the above-mentioned medchem work, and it is restricted to the deeper evaluation of 2 derivatives, namely 3a and 3c, which have been selected on the basis of properties reported in Pharmaceuticals 2023, 16, 1194. The reported effect on amyloid levels is a piece of work that hardly fits here and should have been included in the previously published manuscript. Indeed, the selection of 3a and 3c for in vivo studies is not based (exclusively) on the outcomes reported in Table 2, in which compounds 3l and 3k are the most promising ones.

Response: Indeed, the main rationale for prioritizing some of the compounds for further in vivo investigation is cited repeatedly in the text and relates to their combined anticholinesterase and antioxidant properties reported in a previous publication. The antiamyloidogenic screening was intended and conducted as a complementary investigation on a larger number of compounds, to more fully characterize the representatives of the series that showed a lack of neuronal cytotoxicity. Then again, our main objective was the development of new hybrid molecules with multitarget effects, which requires that compounds with optimal properties combine pleiotropic anti-Alzheimer activities rather than exert a pronounced yet isolated antioxidant, anticholinesterase or anti-amyloidogenic action. The occurrence of а significant overlapping of these effects (that are not necessarily maximal) within the conducted pharmacodynamic studies was imperative to the design of the in vivo experiment.

In fact, we recently reported the beneficial effect of subchronic treatment with the 3c compound in a rat model of pinealectomy + icvAβ1-42 on memory impairment and model-induced increased AD biomarkers Aβ1-42 and pTAU in the hippocampus [Tchekalarova J, Ivanova P, Krushovlieva D, Kortenska L, Angelova VT. Protective Effect of the Novel Melatonin Analogue Containing Donepezil Fragment on Memory Impairment via MT/ERK/CREB Signaling in the Hippocampus in a Rat Model of Pinealectomy and Subsequent Aβ1-42 Infusion. International Journal of Molecular Sciences. 2024; 25(3):1867. https://doi.org/10.3390/ijms25031867]. Experiments with the other most active compound 3a in the above in vivo rat model of Alzheimer's disease are ongoing.

Point # 2 It seems like the authors would like to fit all the data they have collected and were not able to include in the manuscript published in 2023, but this makes the incipit of the result and of the discussion sections in this manuscript not linear by reporting results on 15 compounds that do not have any follow-up and for which we miss their in vitro profile. This also forces the authors to continuously refer to previously published data and discussions on previously published data. Hence, my suggestion is to rearrange the manuscript by only focusing on the 3a and 3c derivatives, without any discussion on the amyloid-lowering properties of compounds that have not been subjected to further investigations in this manuscript. The main text should show any data on amyloid levels of the two compounds, while all the others should be removed because they have no impact on the selection. Introduction, results, and discussion sections should be rearranged consequently.

Response: It is impossible to know the properties of the compounds before conducting the experiment, which is ultimately its purpose. The ranking of compounds in terms of cytotoxicity, for example, is precisely on the basis of the cytotoxicity data of all compounds in the series which should be reported for evidentiary purposes. Similarly, undertaking the Aβ42 measuring experiment, we screened selected compounds that have shown favorable cytotoxicity profiles in the previous study (among all analogues in the series). The same principle is applied to the Aβ42 study, which additionally helps to narrow down lead compounds for the in vivo study. Thus, based on the present results (anti-amyloidogenic properties) as well as those previously reported (on cytotoxicity, anticholinesterase and antioxidant activities), only 2 compounds with multimodal targeting of Alzheimer’s pathogenetic features continued in further animal experiments.  It is however, of evidential value to report the experimental results for all of the screened compounds with low cytotoxicity and not just those that have proved beneficial.

Point # 3 In tables reporting results from all in vivo or ex vivo experiments, it is important to indicate the administered doses of the reference compound, namely donepezil, galantamine, and 3a and 3c. Indeed, it seems that not the same administration dose was used (1mg/kg vs. 35 mg/kg). If this information is not specified and not clearly highlighted in the test, the reader gets a false perception of the relative properties of the two selected derivatives. Doses administered should also be clearly motivated and discussed.

Response: The doses of donepezil, scopolamine and galantamine are according to literature data. The doses of both compounds 3a and 3c were based on acute toxicity data. Doses 1/10 of the LD50 were used for them. Usually, when no data are available on the toxicity of the compounds, fractions of the LD50, 1/5, 1/10, etc., are used for repeated dosing. (Antov, G., Kh, Z., Zh, K., Mikhaĭlova, A., & Shumkov, N. (1992). An experimental evaluation of the acute and chronic oral toxicity of the antibiotic Bactericin for plant protection. Problemi na Khigienata, 17, 109-116.)

In Tables 1 and 2, in which we present the results of the hematological and biochemical analysis of blood from the experimental animals, we have supplemented the repeatedly administered doses of the investigated compounds

Point # 4 Conclusions should be consistent with the data, as discussed above, and clearly highlight the study's limitations in light of the two following points: relative doses administered and the absence of in vivo proof with AD models of anti-amyloidogenic activity.

Response: Thank you for the comment. The conclusions drawn are based solely on the experiments performed. Treatment doses are accurately reported, and it is pointed out repeatedly throughout the text that the anti-amyloid study was conducted in an in vitro setting. As we mentioned in Response to Point #1, we recently reported that the compound 3c showed a promising anti-pathogenic activity in a rat model of pinealectomy + icvAβ1-42 on memory impairment and model-induced elevated AD biomarkers Aβ1-42 and pTAU in the hippocampus that was comparable to the effects of the reference drug melatonin [Tchekalarova J, Ivanova P, Krushovlieva D, Kortenska L, Angelova VT. Protective Effect of the Novel Melatonin Analogue Containing Donepezil Fragment on Memory Impairment via MT/ERK/CREB Signaling in the Hippocampus in a Rat Model of Pinealectomy and Subsequent Aβ1-42 Infusion. International Journal of Molecular Sciences. 2024; 25(3):1867. https://doi.org/10.3390/ijms25031867]. Furthermore, in this in vivo rat model of AD we have found that 3c supplementation facilitated non-amyloidogenic signaling via receptor-related signaling (MT1A and MT2B /ERK/CREB). These earlier findings have also been mentioned in the new version of the conclusion.

Point # 5 Further comments:

The authors should add statistical analysis (e.g., an ANOVA test) to Figure 2 and add the number of repeats to the figure label. FURTHERMORE, the acronym identification of a given compound should be kept consistent. In the Figure 2 label, donepezil is acronymized as DNPZ, but in the bar graph, it is labeled as D. Please be uniform.

Response: Thank you for your remark. Figure 2 and its caption have been corrected and improved. 

Point # 6 Concerning the selection of donepezil and galantamine as reference standards for the anti-amyloidogenic activity reported in Figure 2, it must be underlined that those compounds are not known as anti-aggregating agents but anticholinesterase ones.

Response: Thank you for your note. Both drugs are mentioned as anticholinesterase agents as early as in the introduction of the article. For the sake of reader's ease and clarity, we will additionally mention the lack of antiamyeloidogenic properties of both drugs in the ELISA results section. Based on previous results showing the inhibitory activity of galantamine  on the aggregation of Aβ peptides, as well as on Aβ clearance by microglial phagocytosis, this compound was included as a positive control of inhibition [Takata, K., Kitamura, Y., Saeki, M., Terada, M., Kagitani, S., Kitamura, R., ... & Shimohama, S. (2010). Galantamine-induced amyloid-β clearance mediated via stimulation of microglial nicotinic acetylcholine receptors. Journal of biological chemistry, 285(51), 40180-40191.], [Galantamine inhibits β-amyloid aggregation and cytotoxicity] [Matharu, B., Gibson, G., Parsons, R., Huckerby, T. N., Moore, S. A., Cooper, L. J., ... & Austen, B. (2009). Galantamine inhibits β-amyloid aggregation and cytotoxicity. Journal of the neurological sciences, 280(1-2), 49-58.] Galantamine also inhibits Aβ-mediated cytostatic autophagy by reducing ROS aggregation [Jiang, S. et al. Galantamine inhibits β‐amyloid‐induced cytostatic autophagy in PC 12 cells through decreasing ROS production. 51, e12427 (2018)].

Other studies showed that donepezil and galantamine could promote neuroprotection and reduce Aβ deposition [Moreira, N. C. D. S., Lima, J. E. B. D. F., Marchiori, M. F., Carvalho, I., & Sakamoto-Hojo, E. T. (2022). Neuroprotective effects of cholinesterase inhibitors: current scenario in therapies for Alzheimer’s disease and future perspectives. Journal of Alzheimer's disease reports, 6(1), 177-193].

Point # 7 Also, attention must be paid to the claim of neuroprotective action, which should be limited to experiments in which neuroprotection has been effectively evaluated. In detail, regarding the current discussion of Figure 2, the authors should correct the sentence: "The most pronounced neuroprotective effect was found for the tert-butyl-2-hydrazinyl-2-oxoethylcarbamate derivatives 3k and 3l..." Indeed, Figure 2 does not refer to the neuroprotection activity (no cell viability was assayed) of the tested compounds but to their ability to inhibit the amount of amyloid in the supernatant.

Response: The relationship between Aβ42 levels and neuroprotection is indirect but still very much valid and can be inferred as a logical corollary given the many accumulated data on the neurotoxic effects of Aβ42 species.  [Chen, Gf., Xu, Th., Yan, Y. et al. Amyloid beta: structure, biology and structure-based therapeutic development. Acta Pharmacol Sin 38, 1205–1235 (2017). https://doi.org/10.1038/aps.2017.28;],

[ Hampel, H., Hardy, J., Blennow, K. et al. The Amyloid-β Pathway in Alzheimer’s Disease. Mol Psychiatry 26, 5481–5503 (2021). https://doi.org/10.1038/s41380-021-01249-0; ] [Datki Z, Papp R, Zádori D, Soós K, Fülöp L, Juhász A, Laskay G, Hetényi C, Mihalik E, Zarándi M, Penke B. In vitro model of neurotoxicity of Abeta 1-42 and neuroprotection by a pentapeptide: irreversible events during the first hour. Neurobiol Dis. 2004 Dec;17(3):507-15. doi: 10.1016/j.nbd.2004.08.007;]

[ Karisetty BC, Bhatnagar A, Armour EM, Beaver M, Zhang H, Elefant F. Amyloid-β Peptide Impact on Synaptic Function and Neuroepigenetic Gene Control Reveal New Therapeutic Strategies for Alzheimer's Disease. Front Mol Neurosci. 2020 Nov 13;13:577622. doi: 10.3389/fnmol.2020.577622.]

Point # 8 Please check the correct use of capital letters in  the AChE acronym.

Response: Thank you for your remark. Capital letters in the AChE acronym have been corrected. 

Point # 9 Experimental part: It is important to make readers able to repeat experiments. Hence, all applied procedures are described in detail, specifying volumes, times, buffer composition, and solution concentration. They cannot be general (see section 4.1 and subsections which do not list any detail on stock solution preparation, concentrations, solvents etc)

Response: All the methods used are well known and used for many years in many scientific laboratories. The requirements of most quality journals, such as IJMS, well-established methods to be briefly described and appropriately cited, to avoid plagiarism.

Reviewer 2 Report

Comments and Suggestions for Authors

Review ijms-2977993

Overall this paper contains interesting and important data.  The authors have collected much data regarding the chemical compounds studied.  However, with the massive amount of data presented, the presentation becomes difficult to understand and moves off into tangents that are not hugely relevant to the topic of dementia.  Also, if the main target of this paper is the effect of the different agents on Aβ42 that investigation should have been done with more rigor as noted below.   There needs to be much clarification and improvement in presentation and other issues as noted below.

1-The abstract is far too long and needs editing.

2-This paper and the author’s previous paper give little information on why the specific compounds were chosen.  They note that they were related to melatonin or donepezil but why were the specific 15 compounds chosen?  Was it just that they were easy to synthesize?  What is the numbering scheme?  What is the 3 in front of the letters (3a,3b….) and why are 3f,3g missing?  What does the 6 in front of the letter mean?

3-Could the toxicology data from all three routes be combined into a single larger table along with the LD information---I am not certain that the symptoms are critically important?

4-Regarding the Aβ42 in neuronal SH-SY5Y cells,  could these effects be due to a non-specific toxic effect of the compound tested? Was there any attempt to determine the viability of the cultured cells after exposure?  According to the data in the author’s previous paper the IC50 for cytoxicity in the SH-SY5Y cells is close to the 100uM concentration used to measure the Aβ42.  Because of this issue, it would have been useful to study the effect on Aβ42 at lower concentrations of the test agent say 20uM which is well below the IC50

5-The authors conflate Aβ42 levels with “neuroprotection” in the 3rd and 4th paragraph of page 4.  The authors must be very specific about what they state.  They have not proven the relationship between Aβ42 and “neuroprotection” in this system and model.  In the following paragraph the authors note that compounds 3a and 3c  have antioxidant and AchE activities---what is the reference for this?  I see that this was noted later in the paper—so this should be stated.

6-For the results on the biochemical  testing such as in table 4 what concentration of the agent was used in each of these cases?  How was the p-value “vs reference range” computed?  Most of the changes although some may be statistically significant are not clinically significant.

7-On page 8 the authors state that scopolamine lowered AChE activity but DNPZ did not.  However the result should be opposite since DNPZ is an AChE blocker and scopolamine blocks the muscarinic receptor.  How is this explained?

8-The authors do not correct for multiple testing in their statistical analyses—simply performing a large number of paired comparisons is not appropriate.

Comments on the Quality of English Language

see above

Author Response

Thank you for the careful evaluation of our manuscript. We have revised the manuscript taking into account the suggested modifications. All changes in the MS are highlighted in red.

Point #1 1-The abstract is far too long and needs editing.

Response: Тhank you for the valuable note. The abstract has been revised according to the journal's requirements and updated in the manuscript text.

Point # 2-This paper and the author’s previous paper give little information on why the specific compounds were chosen.  They note that they were related to melatonin or donepezil but why were the specific 15 compounds chosen?  Was it just that they were easy to synthesize?  What is the numbering scheme?  What is the 3 in front of the letters (3a, 3b….) and why are 3f,3g missing?  What does the 6 in front of the letter mean?

Response: Thank you for your comment. In the article, as in our previous article, we justified the choice of hybrid molecules containing fragments of donepezil and melatonin. Our goal is to develop inhibitors that interact with multiple targets simultaneously to restore the impaired neurological functions associated with Alzheimer's disease. In addition, we used molecular docking to aid the design process and verified the in silico pharmacokinetic parameters of the compounds selected for synthesis. As a result, we obtained several families of hybrid molecules with different substitutions. The compound numbers mentioned in the synthetic scheme published in our previous article provide clarity on these synthesised compounds. For better understanding, a generalised synthetic scheme has been included in the Results section of Figure 1.

Point # 3  Could the toxicology data from all three routes be combined into a single larger table along with the LD information---I am not certain that the symptoms are critically important?

Response: All tables concerning tox data are removed.

The following phrases were embedded into the Results section: At 2000 mg/kg, 100% of the experimental animals died, at 1000 mg/kg, 66.7% of the animals died, and at 500 mg/kg, a 33.3% lethality was found. Administration of the low doses of 250 and 125 mg/kg did not cause lethality. Assuming that the lowest lethal dose is 500 mg/kg and the highest non-lethal dose is 250 mg/kg, we can calculate that the average lethal dose for both compounds 3a and 3c is approximately 354 mg

Point #4  Regarding the Aβ42 in neuronal SH-SY5Y cells, could these effects be due to a non-specific toxic effect of the compound tested? Was there any attempt to determine the viability of the cultured cells after exposure?  According to the data in the author’s previous paper the IC50 for cytoxicity in the SH-SY5Y cells is close to the 100uM concentration used to measure the Aβ42.  Because of this issue, it would have been useful to study the effect on Aβ42 at lower concentrations of the test agent say 20uM which is well below the IC50

Response: Thank you for the legitimate and important observation.

As you yourself noticed, the previously reported IC50 data of the compounds on the neuronal SH-SY5Y cells are heterogeneous, but most of them range around 100 M. It should be noted that these half-inhibitory concentrations were estimated at a 72 h exposure time - 3 folds higher compared to the protocol of the ELISA experiment (conducted at a 24 h period of exposure). Given this, we chose a treatment concentration of 100 M to still effectively induce potential changes in cell biology over a shorter period. Furthermore, as can be noticed, there appears to be no correlation between the Aβ42 reducing capacity and the IC50 value of the compounds (e.g., the estimated IC50 of galantamine and donepezil were 79.3 and 300 M, respectively, but the effects of both compounds on Aβ42 levels were similarly negligible). Last but not least, by definition, the ELISA experiment is performed strictly normalized to cell count for each sample, to eliminate variations in protein concentration due to cell density. Therefore, cytotoxicity and variations of cell count can be ruled out as factors in the observed changes in Aβ42 concentration in the cell supernatant.

Point # 5  The authors conflate Aβ42 levels with “neuroprotection” in the 3rd and 4th paragraph of page 4.  The authors must be very specific about what they state.  and They have not proven the relationship between Aβ42 and “neuroprotection” in this system model.  In the following paragraph the authors note that compounds 3a and 3c have antioxidant and AchE activities---what is the reference for this?  I see that this was noted later in the paper—so this should be stated.

Response: The relationship between Aβ42 levels and neuroprotection is indirect but still very much valid and can be inferred as a logical corollary given the many accumulated data on the neurotoxic effects of Aβ42 species. [Chen, Gf., Xu, Th., Yan, Y. et al. Amyloid beta: structure, biology and structure-based therapeutic development. Acta Pharmacol Sin 38, 1205–1235 (2017). https://doi.org/10.1038/aps.2017.28; Hampel, H., Hardy, J., Blennow, K. et al. The Amyloid-β Pathway in Alzheimer’s Disease. Mol Psychiatry 26, 5481–5503 (2021). https://doi.org/10.1038/s41380-021-01249-0; Datki Z, Papp R, Zádori D, Soós K, Fülöp L, Juhász A, Laskay G, Hetényi C, Mihalik E, Zarándi M, Penke B. In vitro model of neurotoxicity of Abeta 1-42 and neuroprotection by a pentapeptide: irreversible events during the first hour. Neurobiol Dis. 2004 Dec;17(3):507-15. doi: 10.1016/j.nbd.2004.08.007; Karisetty BC, Bhatnagar A, Armour EM, Beaver M, Zhang H, Elefant F. Amyloid-β Peptide Impact on Synaptic Function and Neuroepigenetic Gene Control Reveal New Therapeutic Strategies for Alzheimer's Disease. Front Mol Neurosci. 2020 Nov 13;13:577622. doi: 10.3389/fnmol.2020.577622.]

The reference to the antioxidant and anticholinesterase activity of compounds 3a and 3c has been added.

Point # 6  For the results on the biochemical testing such as in table 4 what concentration of the agent was used in each of these cases?  How was the p-value “vs reference range” computed?  Most of the changes although some may be statistically significant are not clinically significant.

Response: The doses of the compounds with which the animals were treated have been added to the tables. You are absolutely right, there is no way to estimate a statistically significant difference with the reference values, and we just wanted to note that our results are outside the stated reference limits.

Point # 7-On page 8 the authors state that scopolamine lowered AChE activity but DNPZ did not.  However the result should be opposite since DNPZ is an AChE blocker and scopolamine blocks the muscarinic receptor.  How is this explained?

Response: In our experimental conditions scopolamine (administered alone) was found to increase the brain AChE activity by 44% compared with the control group (Figure 3). DNPZ, 3a and 3c (administered alone) did not affect the activity of this enzyme. In combination with scopolamine, DNPZ significantly reduced AChE activity by 37%, and 3a and 3c reduced it by 26% and 33%, respectively, compared with the scopolamine-treated group (alone).

Point # 8  The authors do not correct for multiple testing in their statistical analyses—simply performing a large number of paired comparisons is not appropriate.

Response: We agree that the statistical analysis could be more precise, with more comparisons between groups, but for a clearer presentation of the statistical differences compared to the pathological model, we chose this way of presentation.

Round 2

Reviewer 2 Report

Comments and Suggestions for Authors

I would refer the authors to my previous comments which still need addressing in the manuscript.

What is the meaning of the number and letters used to refer to the various compound?  Why were these specific compounds chosen among the large number of possibilities?  The authors still have not explained why there is no 3f, 3g or why some are labellec 6 and som 3?

The authors cannot use the term “neuroprotection” when referring the Ab42 levels. Refer to the levels themselves. 

Regarding the statistics, the reviewer did not ask for more tests just correction for multiple testing.   If the authors do not want to do additional statistical tests, then they can use the Bonferroni correction.  Thus if there are 10 tests then the p value for significance should be 0.005 not 0.05.

The authors have not explained in the paper the anomalous results for DNPZ on ACHE activity.

The manuscript needs a discussion as to why cell death cannot explain the results in figure 2. 

Comments on the Quality of English Language

moderate review still needed

Author Response

Point #1. I would refer the authors to my previous comments which still need addressing in the manuscript.

1.What is the meaning of the number and letters used to refer to the various compound? The authors still have not explained why there is no 3f, 3g or why some are labellec 6 and som 3?

Response: The numbers and letters correspond to the synthetic scheme shown on page 3, Figure 1. "The synthetic scheme and molecular structures of the hybrid hydrazide-hydrazone compounds studied are based on benzylpiperadine (3a-d), pyrrolidine (3e, 3h, 3i), tert-butyl-2-hydrazinyl-2-oxoethylcarbamate derivatives (3j, 3k, 3l), indole (3n, 3o) and vanillyl (3r) fragments, and sulfonyl hydrazones (6a and 6k) [31]".

This is a commonly used method of generically designating the starting compounds as 1 and 2, while the product is designated as 3. The compounds in series 3 are hydrazide hydrazones, while those obtained from reagents 4 and 5 are sulfonylhydrazones numbered 6.

Variations in the substituents - R, R1 and R2 are indicated by lower case letters of the alphabet after the positions 3 and 6. These substituents are visible in the molecular structure formulae of the hybrids studied.

In order to avoid plagiarism, we cannot repeat the complete synthetic scheme and, therefore, we have not included in this work compounds that we have not studied. Compounds 3f and 3g are not included because they are cytotoxic and not suitable for the present study.

Point # 2. Why were these specific compounds chosen among the large number of possibilities?  

Response: Our primary goal is to identify lead compounds that can act as multi-targeting agents for the treatment of Alzheimer's disease. In line with this, we are intensifying our research with 3a and 3c. As we observed in our previous study, these compounds have significant acetylcholinesterase activity and excellent antioxidant properties. They also show minimal cytotoxicity, good blood-brain barrier permeability and significant anti-Aβ activity. These multi-targeting properties are key in our quest for effective treatment of Alzheimer's disease.

On page 5, we have added the following sentence in red to the text:

"We continue to investigate the AChE inhibitors (3a and 3c), which interact with multiple targets simultaneously and show excellent antioxidant activities, minimal cytotoxicity, good blood-brain barrier permeability and significant anti-Aβ activity".

Point # 3 The authors cannot use the term “neuroprotection” when referring the Ab42 levels. Refer to the levels themselves. 

Response: In the interpretation of the Ab42 study results, the term neuroprotection has been replaced by the term reduction in measured neurotoxin levels, as recommended.

Point # 4. Regarding the statistics, the reviewer did not ask for more tests just correction for multiple testing.   If the authors do not want to do additional statistical tests, then they can use the Bonferroni correction.  Thus if there are 10 tests then the p value for significance should be 0.005 not 0.05.

 Response: We fully agree with this comment! We performed a Bonferroni corrected post hoc test and the new p-value for significance in all ex vivo results is p≤ 0.01. 

Point # 5 The authors have not explained in the paper the anomalous results for DNPZ on ACHE activity.

Response: In our opinion, there is no anomaly in the results obtained for donepezil on AChE activity. DNPZ, applied alone to the experimental animals, reduced the activity of the enzyme by only 21% (p= 0.049, post hoc test Bonferroni corrected p= 0.007480294). In animals treated with scopolamine (the in vivo model of Alzheimer's dementia), donepezil reduced enzyme activity by 37% (p=7.36251E-07), Bonferroni corrected (p=1.22708E-07). The results of the two compounds studied are also similar, although they have a weaker effect on enzyme activity compared to donepezil. 3a reduced enzyme activity by 26% and 3c by 33% compared to the scopolamine only group.

Point # 6 The manuscript needs a discussion as to why cell death cannot explain the results in figure 2. 

Response: Cell death (and in this sense the presence of cytotoxicity) is an entirely unrelated effect to the capacity of a compound to lower β amyloid levels. Different cellular mechanisms, signaling pathways, and enzyme systems are involved in the two processes, which do not necessarily intersect.

Round 3

Reviewer 2 Report

Comments and Suggestions for Authors

Two minor comments--Some of the comments the authors wrote to respond to the comments are very important and should be in the paper.  especially the response to point #2. 

In response to point #6 the authors state that the ability to suppress Ab42 is not related to cell death.  However if the cells die they cannot make Ab42 so there must be a relationship.

Comments on the Quality of English Language

improving

Author Response

Reviewer # 3

Point #1 Some of the comments the authors wrote to respond to the comments are very important and should be in the paper. especially the response to point #2.

 Response: We agree to this relevant note and a text was included on p. 3 in the Introduction and also on p. 5 in the Introduction.

Point# 2 In response to point #6 the authors state that the ability to suppress Ab42 is not related to cell death.  However if the cells die they cannot make Ab42 so there must be a relationship.

Response: We agree with this comment. Indeed, there must be a close relationship between Ab42 expression and cell death if the drug has toxic activity and low cell viability. In the present study, the compounds used for the Ab42 assay on SH-SY5Y neuronal cells were selected based on their low toxicity (IC50 values less than 100 µM) reported in our recent study (https://doi.org/ 10.3390/ph16091194). In addition, the two lead compounds 3a and 3c showed IC50 values > 300 µM, suggesting that their efficacy to down-regulate Ab42 at a concentration of 100 µM is not due to potential cell death. This issue was commented in the first paragraph in Discussion section.
